# Less Is More? The Association between Survival and Follow-Up Protocol after Treatment in Oral Cavity Cancer Patients from a Betel Quid-Prevalent Region

**DOI:** 10.3390/ijerph182312596

**Published:** 2021-11-29

**Authors:** Shih-An Liu, Chen-Chi Wang, Rong-San Jiang, Yu-Chi Tung

**Affiliations:** 1Department of Otolaryngology Head & Neck Surgery, Taichung Veterans General Hospital, Taichung 40705, Taiwan; saliu@vghtc.gov.tw (S.-A.L.); ccwang@vghtc.gov.tw (C.-C.W.); rsjiang@vghtc.gov.tw (R.-S.J.); 2Center for Quality Management, Taichung Veterans General Hospital, Taichung 40705, Taiwan; 3Faculty of Medicine, School of Medicine, National Yang Ming Chiao Tung University, Taipei 11221, Taiwan; 4Institute of Health Policy and Management, College of Public Health, National Taiwan University, Taipei 10055, Taiwan; 5Department of Medical Research, Taichung Veterans General Hospital, Taichung 40705, Taiwan

**Keywords:** oral cavity cancer, post-treatment surveillance, choosing wisely

## Abstract

The optimal follow-up protocol after treatment of oral cavity cancer patients is still debatable. We aimed to investigate the impact of frequency of different imaging studies and follow-up visits on the survival of oral cavity cancer patients. The current study retrospectively reviewed oral cavity cancer patients who underwent surgical intervention in our hospital. Basic demographic data, tumor-related features, treatment modalities, imaging studies, and clinic visits were recorded. Cox proportional hazard model was used to examine the influence of variables on the survival of oral cavity cancer patients. In total, 741 patients with newly diagnosed oral cavity cancer were included in the final analysis. Overall, the frequency of imaging studies was not associated with survival in the multivariate analysis, except PET scan (hazard ratio [HR]: 5.30, 95% confidence interval [CI]: 3.57–7.86). However, in late-stage and elder patients, frequent head and neck CT/MRI scan was associated with a better prognosis (HR: 0.55, 95% CI: 0.36–0.84; HR: 0.52, 95% CI: 0.30–0.91, respectively). In conclusion, precision medicine is a global trend nowadays. Different subgroups may need different follow-up protocols. Further prospective study is warranted to clarify the relationship between frequency of image studies and survival of oral cavity cancer patients.

## 1. Introduction

There are three types of quality problems in the healthcare system: underuse, misuse, and overuse. Eradication of overuse is one of the top priorities for most healthcare systems in developed countries as this will reduce healthcare costs and improve patient care [1]. According to a report by the Institute of Medicine, nearly 30% of all medical costs (more than USD 750 billion) per year can be attributed to unnecessary or inefficient services [2]. A previous study in Taiwan found a higher overuse index was associated with higher total medical costs per capita in a population-based analysis [3].

In terms of the management for cancer patients, it is estimated that the total expenditure for cancer care in the United States rose from USD 125 billion in 2010 to USD 173 billion in 2020 [4]. Therefore, the American Society of Clinical Oncology (ASCO) proposed the Top Five Choosing Wisely (CW) list of low-value tests and procedures for cancer patients in 2013 [5]. However, the rates of concordance with the ASCO’s CW in the United States showed room for improvement. Further effort on CW measures is needed to augment patient care and enhance the value of healthcare [6].

Loco-regional recurrence rate in oral cavity cancer patients is relatively high when compared with that of other types of cancers and was reported to affect 19% to 34% of oral cavity cancer patients [7]. Therefore, for head and neck cancer patients, routine follow-up is recommended rather than returning only when symptoms reappear [8]. However, recent studies have indicated that there is little benefit of follow-up in asymptomatic head and neck cancer patients [9]. With respect to image surveillance, a study from Italy showed clinical and radiologic follow-up in head and neck cancer patients could detect more recurrences or second primary cancers than symptom-driven monitoring. Nevertheless, the prognosis remained similar between these two groups [10]. A single institute study from the United States on 46 oral cavity cancer patients found routine head and neck MRI surveillance may be unnecessary and costly as only 1 out of 46 (2.2%) had a true positive regional recurrence [11].

Many studies on head and neck cancer surveillance concluded that the evidence is still weak in the guideline recommendations with respect to test interval and duration, and that further investigation is warranted [9,10,12,13]. As most failures are within the first 2 years after treatment, frequent follow-up is suggested at that time period [9]. A previous study proposed follow-up plan for head and neck cancer patients after comprehensive treatment as below: physical examination of head and neck region along with endoscopy examination every 3 months for the first year, every 4–6 months from the second to fourth year, then once in the fifth year. Head and neck MRI of CT scan at least twice a year within 2 years after treatment and once a year in the following years [10]. However, according to the guidelines of National Comprehensive Cancer Network (NCCN), there are no consensus on the frequency and protocol of routine surveillance imaging in the asymptomatic patients and the practice differs widely among healthcare institutes [14]. Nevertheless, most of the abovementioned studies were conducted in Western countries. The etiology of oral cavity cancer in Western countries is different from that in Eastern countries, where betel quid chewing tends to be the main cause [15]. There is no study on surveillance of oral cavity cancer patients in a betel quid-prevalent area. Thus, we aimed to investigate the relationship between prognosis and follow-up protocol after comprehensive treatment in oral cavity cancer patients.

## 2. Materials and Methods

This was a single institute study, and the study design was a retrospective chart review. The current study was reviewed and approved by the Institutional Review Board (IRB) of Taichung Veterans General Hospital (TCVGH) (date: 15 April 2021, approval number: CE21097B). Informed consent was exempted by the IRB of TCVGH as the treatment had been completed and no interventional procedure was given.

### 2.1. Data Sources, Treatment and Follow-Up Protocol

The current study reviewed over 1000 medical charts of oral cavity cancer patients admitted for surgical intervention in the TCVGH from January 2011 to December 2020 with the observation ending on 31 August 2021. Therapeutic protocols for all patients were reviewed by multidisciplinary meeting and were conducted in accordance with the consensus guideline of the oral cancer team of TCVGH. The treatment and follow-up protocol for oral cavity cancer patients are mainly derived from NCCN guidelines. In brief, surgical intervention is suggested for the first line therapeutic modality followed by radiotherapy with/without chemotherapy if adverse histological features presented after pathological examination. Patients was asked to go back to clinic for detailed physical examination of head and neck region every month within the first year followed by every 2–4 months in the second to fifth year years. Surveillance imaging was arranged twice a year during the first 2 years followed by once a year after 2 years.

### 2.2. Independent/Dependent Variables

The variables for analysis included age, gender, personal habits, tumor stage, treatment modalities, tumor location, histological features (such as perineural invasion, lymph-vascular invasion, and extra-nodal extension). Those who smoked cigarettes, drank alcohol, or chewed betel quid only on special occasions such as wedding banquets, family reunions, or birthday parties were not considered habitual users. This study calculated the numbers of imaging studies (including computerized tomography (CT), magnetic resonance imaging (MRI), liver sonogram, whole body bone scan, and positron emission tomography (PET)) after comprehensive treatment, which were arranged because of oral cavity cancer. In addition, the numbers of clinic visits were limited to those attending follow-up for oral cavity cancer. Local recurrence was proved by pathological examination in all cases. Distant metastasis was assessed mainly by imaging study. Survival duration was defined as the period from the date of surgery to the date of death because of oral cavity cancer or the last date of follow-up in the study.

### 2.3. Statistical Analysis

Descriptive statistics was used to present the demographic data of our patients. In addition, the differences in continuous variables between subgroups were compared using Student’s *t*-test while nominal or ordinal variables were analyzed using the chi-square test. Furthermore, factors that could influence the survival period were examined by the Cox proportional hazard model. For ease of analysis, we divided our population into subgroups according to median or mean. In addition, we pooled all variables into a Cox proportional hazard model first and then gradually excluded variables which were not statistically significant. All analyses were computed by SPSS for Windows, version 22.0 (SPSS, IBM Corp, Chicago, IL, USA). Statistically significant level was set at *p* < 0.05.

## 3. Results

### 3.1. Study Population

From January 2011 to December 2020, a total of 1045 patients with oral cavity cancer were admitted for surgical intervention. Among them, 251 patients (24.0%) underwent operation due to recurrent disease. In addition, 21 patients (2.0%) had a histological report other than squamous cell carcinoma, and 32 patients (3.1%) were lost to follow-up. Finally, there were 741 patients (71.7%) who underwent surgical intervention as their main therapeutic method and further analysis of these patients’ data was conducted. The average age at diagnosis was 54.8 (+/−15.5) years and the majority of the patients were male (*n* = 655, 88.4%). The most prevalent primary site was the tongue (*n* = 338, 45.6%). All patients were restaged in accordance with the American Joint Committee on Cancer (8th edition). Based on pathological stage, 235 patients (31.7%) had stage 1 disease, while 156 patients (21.1%), 114 patients (15.4%), and 236 patients (31.8%) had stage 2, stage 3, and stage 4 disease, respectively. As for the patients’ pathological features, 198 patients (26.7%) had perineural invasion and 141 patients (19.0%) had angiolymphatic involvement. Moreover, extra-nodal extension was found in 92 patients (12.4%). More than half of the patients (*n* = 389, 52.5%) received post-operative radiotherapy with/without chemotherapy. The average follow-up period was 47.9 (+/−35.6) months and 166 patients had died because of oral cavity cancer by the end of the follow-up period. Loco-regional recurrence was noted in 233 patients (31.4%), whereas second primary and distant metastasis was found in 136 patients (18.4%) and 25 patients (3.4%), respectively. The average numbers of imaging studies performed after comprehensive treatment of oral cavity cancer during the follow-up period were as follows: CT: 4.43 ± 3.90; MRI: 0.57 ± 1.19; liver sonogram: 2.88 ± 3.12; whole body bone scan: 2.31 ± 2.68; PET: 1.59 ± 1.87. For ease of further analysis, CT and MRI imaging studies were combined for evaluation of loco-regional recurrence and the average numbers of CT/MRI scan performed after treatment during follow-up period was 5.00 ± 4.15. Among those with loco-regional recurrence, 46 patients (19.7%) were asymptomatic and detected by CT/MRI scan. The average clinic visit during the follow-up period was 42.3 ± 82.7 visits.

### 3.2. Subgroups Analysis

When we stratified our patients based on the survival status, there were no significant differences between these two groups in primary site of tumor, diabetes mellitus, hypertension, second primary, and average clinic visit. However, there were significant difference between these two groups in gender, age, personal habits, tumor stage, pathological features, post-operative radiotherapy, recurrence, distant metastasis, average numbers of imaging studies during the follow-up period, and average follow-up duration. Detailed data are shown in Table 1.

Subgroup analysis was made based on the loco-regional recurrent status. There was no statistical significance between these two groups in the type of surgery in primary tumor. However, there were significant differences between these two groups in type of neck dissection and adjuvant treatment. Detailed data are presented in Table 2.

As the follow-up duration was different between the survival group and the deceased group, the numbers of imaging studies that the patients underwent every year were calculated for further analysis (the numbers of imaging studies divided by the follow-up period [in years]). Additionally, the clinic visits every year were computed in Cox proportional hazard model analysis. We used the average as the cut-off points for continuous variables in order to separate the patients into two groups. That is, age: <54 years vs. >=54 years; CT/MRI: <1.3 times/year vs. >=1.3 times/year; Liver sonogram: <0.66 times/year vs. >=0.66 times/year; whole body bone scan: <0.5 times/year vs. >=0.5 times/year; PET: <0.5 times/year vs. >=0.5 times/year; and clinic visit: <11 visits/year vs. >=11 visit/year. Cox proportional model was used to estimate the hazard ratio of variables that could influence survival. First, all of the variables were put into the Cox proportional hazard model for examination. We excluded the variables that were not significant in the Cox proportional hazard model analysis. Finally, we included pathological stage, pathological features, imaging studies, and clinic visits into the final model. Additionally, age and gender were added for control. The results are presented in detail in Table 3. During the univariate analysis, all variables were significant factors associated with survival of oral cavity cancer patients after comprehensive treatment. However, only age, pathological stage, peri-neural invasion, lymph-vascular invasion, PET, and clinic visits were significant prognostic factors in the multivariate analysis.

We then divided our patients according to pathological stage (early-stage, late-stage) and age (younger than 54 years, 54 years or older), further Cox proportional models using the same variables were performed. In early-stage patients, age, lymph-vascular invasion, and PET were significant prognostic factors in the multivariate analysis (Table 4). Moreover, lymph-vascular invasion, CT/MRI, PET, and clinic visit were significant prognostic factors in the multivariate analysis in late-stage patients (Table 5). In younger patients, peri-neural invasion, lymph-vascular invasion, and PET were significant prognostic factors in the multivariate analysis (Table 6). Furthermore, CT/MRI, liver sonogram, PET, and clinic visits were significant prognostic factors in the multivariate analysis of elder patients (Table 7). Our study found most of the patients with second primary were 54 years or older in the current study (>=54 years: 85 out of 393, 21.6% vs. <54 years: 51 out of 348, 14.7%, respectively, *p* = 0.019). In addition, most of the loco-regional recurrence was found in late-stage patients (early-stage: 87 out of 391, 22.3% vs. late-stage: 146 out of 350, 41.7%, respectively, *p* < 0.001)

Regarding the salvage treatments performed in 233 relapsing patients, near half of them underwent surgery plus chemotherapy (*n* = 113, 48.5%), followed by surgery plus radiotherapy (*n* = 66, 28.3%), and chemo-radiotherapy (*n* = 39, 16.7%). Twelve patients received chemotherapy only (5.2% while 3 patients underwent surgery only (1.3%).

## 4. Discussion

This is the first study on post-treatment surveillance of oral cavity cancer patients from a betel quid-prevalent region. Our study found the frequency of imaging studies, with the exception of PET scan, was not associated with the prognosis of oral cavity cancer patients after comprehensive treatment. In the current study, oral cavity cancer patients receiving frequent PET scans had a poor prognosis when compared to those receiving PET scans less frequently. There was no study addressing the relationship between the frequency of PET scan and survival of oral cavity cancer patients. Only one previous study found a negative PET scan done at the 3- to 6-month window was associated with a better survival at 2 years [16]. In Taiwan, National Health Insurance covers PET scans only when there is a suspicion of malignant tumor recurrence. Moreover, PET scan cannot be used as an initial staging or routine follow-up study. This may explain why PET scan was related to a worse outcome in our population. Previous studies found that routine image surveillance after treatment did not improve survival of gastric and pancreatic cancer [17], renal cell carcinoma [18], lung cancer [19], colorectal cancer [20], as well as head and neck cancer [9,10,13]. The result of the current study was comparable with the abovementioned reports. Our result indicated that patients underwent neck dissection especially bilateral neck dissection had a higher rate of local recurrence. In addition, patients received adjuvant treatment especially chemo-radiotherapy had a higher possibility of local recurrence. Patients receiving neck dissection were considered to have an advanced clinical stage. Moreover, patients with advanced stage or adverse pathological feature would undergo adjuvant treatment. This could explain why patients undergoing neck dissection and adjuvant treatment have a higher loco-regional recurrence rate.

However, our analysis of late-stage patients, found frequent CT/MRI could reduce the risk of death. In addition, more CT/MRI scans was associated with a better outcome as well as a greater number of liver sonograms in elder patients. Borsetto et al., found that recurrence was significantly higher for late-stage than for early-stage head and neck patients in the first year after surgery (20.4% versus 10.0%; *p* < 0.01), especially in oral cavity cancer patients [21]. Another study indicated that clinical and radiologic surveillance detected more recurrence/second primary cancer than symptom-driven monitoring [10]. This might be the reason why more imaging studies could improve outcomes in late-stage oral cavity cancer patients in our study. In addition, it was estimated that 23% of head and neck cancer patients died of a second primary [9]. As most of the patients with second primary were 54 years or older in the current study, this probably explains why frequent CT/MRI scans and liver sonograms might detect second primary cancers earlier and may reduce the risk of mortality.

In the current study, most of the loco-regional recurrence was symptom-driven (*n* = 187, 80.3%), while 46 patients (19.7%) were found during routine imaging study. Imbimbo et al., in their study on radiological follow-up of head and neck cancer patients found that nearly 70% of recurrences were clinically and/or radiologically discovered [10]. The differences might be explained by the fact that our study included all stages of oral cavity cancer patients, whereas the abovementioned study included late-stage head and neck cancer patients. In the current study, most of the loco-regional recurrence was found in late-stage patients. As there was still a certain portion of loco-regional recurrence detected by routine image study, this might be another reason why frequent CT/MRI scans could improve the prognosis of late-stage oral cavity cancer patients.

Previous studies found that there was no statistical significance between the frequency of clinic visits and survival in gastric cancer [22], breast cancer [9], as well as head and neck cancer [9,13]. Our study found that frequent clinic visits were correlated with a worse outcome. The possible explanation might be due to different kinds of cancer patients were included for analysis in varies studies. In addition, diverse healthcare systems and different cultural backgrounds might also be the reasons. It was reported that there was no survival benefit in specialist-led follow-up when compared with non-specialist-led follow-up in breast cancer, colorectal cancer, endometrial cancer, ovarian cancer, cervical cancer, melanoma cancer, and esophageal cancer patients [23]. As all our patients were followed up at our clinic, no such data could be compared. Although we derived our follow-up protocol from the NCCN guidelines, a previous study demonstrated that routine follow up after 3 years is questionable as recurrent disease beyond 3 years was very rare [24].

The limitations of the current study were as follows. First, this was a single institute study, and the external validity of our findings was insufficient. Second, the study design was retrospective, and therefore it was not bias-free. Thirdly, the recruitment period was long and the average follow-up varied greatly among patients. Lastly, although the therapeutic guidelines are standardized in our institute, individual differences among surgeons were uncontrolled.

## 5. Conclusions

According to the results of the current study, we recommend that routine image surveillance should be arranged in late-stage and elder oral cavity cancer patients. However, the optimal follow-up protocol and frequency of image surveillance for oral cavity cancer patients is still debatable. Further prospective study is warranted to clarify the relationship between frequency of imaging studies and survival of oral cavity cancer patients. Further comparative studies are also crucial to compare the efficacy of different protocols.

## Figures and Tables

**Table 1 ijerph-18-12596-t001:** Descriptive statistics of oral cavity cancer patients based on survival status.

Variables	Total No. of Patients (% in Column) (*n* = 741)	No. of Patients (%)		*p* Value
		Survival Group (*n* = 575)	Deceased Group (*n* = 166)	
Gender				0.016
Female	86 (11.6%)	76 (88.4%)	10 (11.6%)	
Male	655 (88.4%)	499 (76.2%)	156 (23.8%)	
Age (years)	54.8 ± 10.5	55.2 ± 10.5	53.2 ± 10.5	0.029
Primary subsites				0.676
Lip	35 (4.7%)	27 (77.1%)	8 (22.9%)	
Gum	71 (9.6%)	56 (78.9%)	15 (21.1%)	
Mouth floor	27 (3.6%)	19 (70.4%)	8 (29.6%)	
Tongue	338 (45.6%)	271 (80.2%)	67 (19.8%)	
Buccal	193 (26.0%)	146 (75.6%)	47 (24.4%)	
Palate	37 (5.0%)	28 (75.7%)	9 (24.3%)	
Retromolar trigone	40 (5.4%)	28 (70.0%)	12 (30.0%)	
Smoking				0.037
No	142 (19.2%)	120 (84.5%)	22 (15.5%)	
Yes	599 (80.8%)	455 (76.0%)	144 (24.0%)	
Pack-year	27.1 ± 23.7	27.2 ± 23.8	27.1 ± 23.3	0.968
Alcohol consumption				<0.001
No	242 (32.7%)	204 (84.3%)	38 (15.7%)	
Social	329 (44.4%)	259 (78.7%)	70 (21.3%)	
Heavy	170 (22.9%)	112 (65.9%)	58 (34.1%)	
Betel quid chewing				<0.001
No	187 (25.2%)	163 (87.2%)	24 (12.8%)	
Yes	554 (74.8%)	412 (74.4%)	142 (25.6%)	
Quid-year	482 ± 623	468 ± 653	529 ± 501	0.261
Diabetes mellitus				0.350
No	618 (83.4%)	484 (78.3%)	134 (21.7%)	
Yes	123 (16.6%)	91 (74.0%)	32 (26.0%)	
Hypertension				0.359
No	519 (70.0%)	408 (78.6%)	111 (21.4%)	
Yes	222 (30.0%)	167 (75.2%)	55 (24.8%)	
Stage				<0.001
Stage I and II	391 (52.8%)	327 (83.6%)	64 (16.4%)	
Stage III and IV	350 (47.2%)	248 (70.9%)	102 (29.1%)	
Peri-neural invasion				<0.001
No	543 (73.3%)	442 (81.4%)	101 (18.6%)	
Yes	198 (26.7%)	133 (67.2%)	65 (32.8%)	
Lympho-vascular invasion				<0.001
No	600 (81.0%)	487 (81.2%)	113 (18.8%)	
Yes	141 (19.0%)	88 (62.4%)	53 (37.6%)	
Extra-nodal extension				0.004
No	649 (87.6%)	515 (79.4%)	134 (20.6%)	
Yes	92 (12.4%)	60 (65.2%)	32 (34.8%)	
Post-operative radiotherapy				<0.001
No	352 (47.5%)	296 (84.1%)	56 (15.9%)	
Yes	389 (52.5%)	279 (71.7%)	110 (28.3%)	
Recurrence				<0.001
No	508 (68.6%)	450 (88.6%)	58 (11.4%)	
Yes	233 (31.4%)	125 (53.6%)	108 (46.4%)	
Second primary				0.490
No	605 (81.6%)	473 (78.2%)	132 (21.8%)	
Yes	136 (18.4%)	102 (75.0%)	34 (25.0%)	
Distant metastasis				<0.001
No	716 (96.6%)	575 (80.3%)	141 (19.7%)	
Yes	25 (3.4%)	0 (0%)	25 (100.0%)	
CT/MRI	5.0 ± 4.2	5.26 ± 4.21	4.12 ± 3.85	0.002
Liver sonogram	2.9 ± 3.1	3.14 ± 3.27	2.01 ± 2.32	<0.001
Bone scan	2.3 ± 2.7	2.57 ± 2.81	1.43 ± 1.93	<0.001
PET scan	1.6 ± 1.9	1.28 ± 1.65	2.67 ± 2.15	<0.001
Clinic visit	42.3 ± 82.7	40.0 ± 37.7	50.1 ± 160	0.420
Follow-up duration (months)	47.9 ± 35.6	50.7 ± 36.0	38.2 ± 32.5	<0.001

Abbreviation: CT/MRI, computerized tomography/magnetic resonance imaging; PET, positron emission tomography. Note: Average value +/− standard deviation are used for continuous variables.

**Table 2 ijerph-18-12596-t002:** Descriptive statistics of treatment modalities of oral cavity cancer patients based on recurrent status.

Variables	Total No. of Patients (% in Column) (*n* = 741)	No. of Patients (%)		*p* Value
		With Recurrence (*n* = 233)	Without Recurrence (*n* = 508)	
Primary				0.090
Transoral	501 (67.6%)	147 (29.3%)	354 (70.7%)	
Compartmental	240 (32.4%)	86 (35.8%)	154 (64.2%)	
Neck dissection				0.031
No	120 (16.2%)	32 (26.7%)	88 (73.3%)	
Unilateral	494 (66.7%)	149 (30.2%)	345 (69.8%)	
Bilateral	127 (17.1%)	52 (40.9%)	75 (59.1%)	
Adjuvant treatment				<0.001
No	352 (47.5%)	86 (24.4%)	266 (75.6%)	
Radiotherapy only	137 (18.5%)	45 (32.8%)	92 (67.2%)	
Chemo-radiotherapy	252 (34.0%)	102 (40.5%)	150 (59.5%)	

**Table 3 ijerph-18-12596-t003:** Factors associated with disease-specific survival in oral cavity cancer patients.

Variables	No. of Patients (*n* = 741)	Univariate		Multivariate	
		HR (95% CI)		HR (95% CI)	
Gender					
Female	86	1.00	(ref)	1.00	(ref)
Male	655	1.81	(0.95–3.43)	1.37	(0.71–2.65)
Age					
<54 years	348	1.00	(ref)	1.00	(ref)
>=54 years	393	0.99	(0.73–1.34)	1.40	(1.01–1.93)
Pathological stage					
Stage I and II	391	1.00	(ref)	1.00	(ref)
Stage III and IV	350	2.67	(1.96–3.67)	1.49	(1.01–2.18)
Perineural invasion					
No	543	1.00	(ref)	1.00	(ref)
Yes	198	2.37	(1.73–3.24)	1.49	(1.06–2.10)
Lymph-vascular invasion					
No	600	1.00	(ref)	1.00	(ref)
Yes	141	2.60	(1.87–3.60)	1.85	(1.25–2.73)
Extra-nodal extension					
No	649	1.00	(ref)	1.00	(ref)
Yes	92	2.45	(1.66–3.61)	1.20	(0.77–1.88)
CT/MRI					
<1.3 times/year	387	1.00	(ref)	1.00	(ref)
>=1.3 times/year	354	1.54	(1.13–2.10)	0.72	(0.51–1.02)
Liver sonogram					
<0.66 times/year	386	1.00	(ref)	1.00	(ref)
>=0.66 times/year	355	0.55	(0.40–0.75)	0.87	(0.59–1.27)
Whole body bone scan					
<0.50 times/year	401	1.00	(ref)	1.00	(ref)
>=0.50 times/year	340	0.53	(0.39–0.73)	0.75	(0.51–1.10)
PET scan					
<0.50 times/year	467	1.00	(ref)	1.00	(ref)
>=0.50 times/year	274	6.76	(4.81–9.50)	5.30	(3.57–7.86)
Clinic visit					
<11 visits/year	443	1.00	(ref)	1.00	(ref)
>=11 visits/year	298	2.78	(2.04–3.79)	1.84	(1.29–2.64)

Abbreviations: CT/MRI, computerized tomography/magnetic resonance imaging; PET, positron emission tomography; HR, hazard ratio; CI, confidence interval.

**Table 4 ijerph-18-12596-t004:** Factors associated with disease-specific survival in early-stage oral cavity cancer patients.

Variables	No. of Patients (*n* = 391)	Univariate		Multivariate	
		HR (95% CI)		HR (95% CI)	
Gender					
Female	47	1.00	(ref)	1.00	(ref)
Male	344	5.49	(0.76–39.83)	4.05	(0.55–29.75)
Age					
<54 years	185	1.00	(ref)	1.00	(ref)
>=54 years	206	1.00	(0.61–1.64)	1.80	(1.06–3.07)
Perineural invasion					
No	336	1.00	(ref)	1.00	(ref)
Yes	55	2.15	(1.23–3.74)	1.55	(0.86–2.79)
Lymph-vascular invasion					
No	377	1.00	(ref)	1.00	(ref)
Yes	14	2.38	(0.95–6.00)	3.15	(1.21–8.24)
CT/MRI					
<1.3 times/year	239	1.00	(ref)	1.00	(ref)
>=1.3 times/year	152	1.87	(1.14–3.05)	1.25	(0.71–2.22)
Liver sonogram					
<0.66 times/year	194	1.00	(ref)	1.00	(ref)
>=0.66 times/year	197	0.73	(0.45–1.19)	0.65	(0.32–1.32)
Whole body bone scan					
<0.50 times/year	201	1.00	(ref)	1.00	(ref)
>=0.50 times/year	190	0.79	(0.48–1.29)	1.11	(0.56–2.19)
PET scan					
<0.50 times/year	290	1.00	(ref)	1.00	(ref)
>=0.50 times/year	101	5.09	(3.10–8.35)	4.73	(2.73–8.18)
Clinic visit					
<11 visits/year	266	1.00	(ref)	1.00	(ref)
>=11 visits/year	125	2.18	(1.33–3.57)	1.67	(0.97–2.88)

Abbreviations: CT/MRI, computerized tomography/magnetic resonance imaging; PET, positron emission tomography; HR, hazard ratio; CI, confidence interval.

**Table 5 ijerph-18-12596-t005:** Factors associated with disease-specific survival in late-stage oral cavity cancer patients.

Variables	No. of Patients (*n* = 350)	Univariate		Multivariate	
		HR (95% CI)		HR (95% CI)	
Gender					
Female	39	1.00	(ref)	1.00	(ref)
Male	311	1.35	(0.68–2.68)	0.99	(0.48–2.05)
Age					
<54 years	163	1.00	(ref)	1.00	(ref)
>=54 years	187	0.93	(0.63–1.37)	1.21	(0.80–1.83)
Perineural invasion					
No	207	1.00	(ref)	1.00	(ref)
Yes	143	1.72	(1.16–2.55)	1.20	(0.78–1.85)
Lymph-vascular invasion					
No	223	1.00	(ref)	1.00	(ref)
Yes	127	1.65	(1.12–2.44)	1.59	(1.05–2.43)
Extra-nodal extension					
No	258	1.00	(ref)	1.00	(ref)
Yes	92	1.61	(1.06–2.45)	1.38	(0.87–2.18)
CT/MRI					
<1.3 times/year	148	1.00	(ref)	1.00	(ref)
>=1.3 times/year	202	1.02	(0.69–1.52)	0.55	(0.36–0.84)
Liver sonogram					
<0.66 times/year	192	1.00	(ref)	1.00	(ref)
>=0.66 times/year	158	0.49	(0.33–0.74)	0.87	(0.55–1.37)
Whole body bone scan					
<0.50 times/year	200	1.00	(ref)	1.00	(ref)
>=0.50 times/year	150	0.41	(0.27–0.62)	0.62	(0.38–1.00)
PET scan					
<0.50 times/year	177	1.00	(ref)	1.00	(ref)
>=0.50 times/year	173	6.45	(3.90–10.66)	5.15	(2.91–9.12)
Clinic visit					
<11 visits/year	177	1.00	(ref)	1.00	(ref)
>=11 visits/year	173	2.55	(1.69–3.85)	1.94	(1.21–3.12)

Abbreviations: CT/MRI, computerized tomography/magnetic resonance imaging; PET, positron emission tomography; HR, hazard ratio; CI, confidence interval.

**Table 6 ijerph-18-12596-t006:** Factors associated with disease-specific survival in younger oral cavity cancer patients.

Variables	No. of Patients (*n* = 348)	Univariate		Multivariate	
		HR (95% CI)		HR (95% CI)	
Gender					
Female	28	1.00	(ref)	1.00	(ref)
Male	320	1.97	(0.62–6.24)	1.20	(0.36–3.95)
Pathological stage					
Stage I and II	185	1.00	(ref)	1.00	(ref)
Stage III and IV	163	2.59	(1.69–3.97)	1.37	(0.81–2.34)
Perineural invasion					
No	247	1.00	(ref)	1.00	(ref)
Yes	101	3.03	(2.00–4.60)	2.09	(1.33–3.29)
Lymph-vascular invasion					
No	276	1.00	(ref)	1.00	(ref)
Yes	72	3.61	(2.34–5.57)	2.29	(1.32–3.96)
Extra-nodal extension					
No	305	1.00	(ref)	1.00	(ref)
Yes	43	2.76	(1.62–4.71)	1.19	(0.64–2.21)
CT/MRI					
<1.3 times/year	170	1.00	(ref)	1.00	(ref)
>=1.3 times/year	178	1.76	(1.15–2.69)	0.91	(0.57–1.45)
Liver sonogram					
<0.66 times/year	175	1.00	(ref)	1.00	(ref)
>=0.66 times/year	173	0.69	(0.46–1.05)	1.42	(0.83–2.44)
Whole body bone scan					
<0.50 times/year	185	1.00	(ref)	1.00	(ref)
>=0.50 times/year	163	0.46	(0.30–0.71)	0.69	(0.39–1.22)
PET scan					
<0.50 times/year	202	1.00	(ref)	1.00	(ref)
>=0.50 times/year	146	7.82	(4.76–12.87)	6.30	(3.52–11.30)
Clinic visit					
<11 visits/year	195	1.00	(ref)	1.00	(ref)
>=11 visits/year	153	2.43	(1.58–3.74)	1.32	(0.81–2.14)

Abbreviations: CT/MRI, computerized tomography/magnetic resonance imaging; PET, positron emission tomography; HR, hazard ratio; CI, confidence interval.

**Table 7 ijerph-18-12596-t007:** Factors associated with disease-specific survival in elder oral cavity cancer patients.

Variables	No. of Patients (*n* = 393)	Univariate		Multivariate	
		HR (95% CI)		HR (95% CI)	
Gender					
Female	58	1.00	(ref)	1.00	(ref)
Male	335	1.74	(0.80–3.80)	1.25	(0.56–2.78)
Pathological stage					
Stage I and II	206	1.00	(ref)	1.00	(ref)
Stage III and IV	187	2.82	(1.76–4.53)	1.53	(0.86–2.72)
Perineural invasion					
No	296	1.00	(ref)	1.00	(ref)
Yes	97	1.72	(1.05–2.84)	0.97	(0.55–1.72)
Lymph-vascular invasion					
No	324	1.00	(ref)	1.00	(ref)
Yes	69	1.73	(1.03–2.90)	1.39	(0.77–2.51)
Extra-nodal extension					
No	344	1.00	(ref)	1.00	(ref)
Yes	49	2.18	(1.24–3.83)	1.61	(0.82–3.14)
CT/MRI					
<1.3 times/year	217	1.00	(ref)	1.00	(ref)
>=1.3 times/year	176	1.34	(0.85–2.12)	0.52	(0.30–0.91)
Liver sonogram					
<0.66 times/year	211	1.00	(ref)	1.00	(ref)
>=0.66 times/year	182	0.42	(0.26–0.67)	0.56	(0.32–0.98)
Whole body bone scan					
<0.50 times/year	216	1.00	(ref)	1.00	(ref)
>=0.50 times/year	177	0.63	(0.40–1.00)	0.78	(0.45–1.36)
PET scan					
<0.50 times/year	265	1.00	(ref)	1.00	(ref)
>=0.50 times/year	128	6.26	(3.85–10.17)	4.85	(2.69–8.75)
Clinic visit					
<11 visits/year	248	1.00	(ref)	1.00	(ref)
>=11 visits/year	145	3.35	(2.13–5.29)	2.60	(1.49–4.54)

Abbreviations: CT/MRI, computerized tomography/magnetic resonance imaging; PET, positron emission tomography; HR, hazard ratio; CI, confidence interval.

## Data Availability

Data is contained within the article.

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
