# Peer review of "Less Is More? The Association between Survival and Follow-Up Protocol after Treatment in Oral Cavity Cancer Patients from a Betel Quid-Prevalent Region"

_ijerph, 2021, doi:10.3390/ijerph182312596_

Round 1

Reviewer 1 Report

The authors have presented a well-designed study for investigating the relationship between oral cavity cancer prognosis and follow-up protocol. I have no comments related to the study. However, i would kindly ask the authors to refrain from using the word "we" too much.

Author Response

Reviewer 1

The authors have presented a well-designed study for investigating the relationship between oral cavity cancer prognosis and follow-up protocol. I have no comments related to the study. However, i would kindly ask the authors to refrain from using the word "we" too much.

Reply: Thanks for your compliment. We have refrained the manuscript and the usage of “we” was reduced from 20 to 10 times.

The introduction lacks a focus on different recommended clinical/imaging follow-up protocols which I would suggest the authors to incorporate and elaborate as that is the aim of the study. As so far the intro is more specified towards the economical side and region-based differences.

Reply: Thanks for your suggestion. We will add pervious recommendation and NCCN guidelines for follow-up image in revised manuscript.

The methodology lacks the description of the adopted "treatment and follow-up schedule/protocol"

Reply: Thanks for your suggestion. We will add more description in the treatment and follow-up protocol in the method section.

Summarize only the key descriptive findings as the reader would lose focus if too much repetition of the data already present in table is done.

Reply: Thanks for your reminding. We’ll focus on the relevant variables in revised manuscript.

The discussion is too general and lacks comparison of the follow-up protocols with the one applied in the study. I would suggest the authors to add this aspect as such to provide more clinically oriented information to the readers. So it is easier to know when and what protocol should be followed at what time point for scanning and clinical follow up in contrast to the westernized protocols.

Reply: Thanks for your suggestion. We’ll add this article as reference and discuss the different aspect of follow-up protocol.

This is not clear why frequent PET was related to worse outcome. Kindly clarify and provide reference

Reply: Thanks for your reminding. We’ll add more description in the relationship between frequent PET and survival.

Elaborate. not clear.

Reply: Sorry for the ambitious. We’ll rewrite the sentence in revised manuscript.

I would suggest to add "comparative studies" as such to compare the efficacy of different protocols.

Reply: Thanks for your suggestion. We’ll add that sentence in revised manuscript.

Reviewer 2 Report

The authors addressed an important issue in the management of OSCC. The study results suggested that the frequency of imaging studies was not associated with survival, except in those with late-stage and old age. Overall, this is a well-written manuscript. There are only some points that need to be clarified.

  1. Because of the long study duration (2011~2020), what edition of the AJCC staging system was used in this study to define the cancer stage?
  2. Kindly suggest that there should be definition of personal habits, including the cigarette smoking, alcohol consumption, and betel quid chewing, in the materials and methods.
  3. In Cox proportional model, the DSS was considered as the end point. However, in the section Independent/dependent Variables, the “date of death” seems included all cause mortality. May need more clarification in the definition of DSS.

Author Response

  1. Because of the long study duration (2011~2020), what edition of the AJCC staging system was used in this study to define the cancer stage?

Reply: Thanks for your reminding. We restaged all the patients in accordance to the American Joint Committee on Cancer (8th edition). We’ll add more description in revised manuscript.

  1. Kindly suggest that there should be definition of personal habits, including the cigarette smoking, alcohol consumption, and betel quid chewing, in the materials and methods.

Reply: Thanks for your reminding. Those who smoked cigarettes, drank alcohol, or chewed betel quid only on special occasions such as wedding banquets, family reunions, or birthday parties were not considered habitual users. We’ll add more description in revised manuscript.

  1. In Cox proportional model, the DSS was considered as the end point. However, in the section Independent/dependent Variables, the “date of death” seems included all cause mortality. May need more clarification in the definition of DSS.

Reply: Thanks for your reminding. It should be the date of death due to oral cavity cancer. We’ll add more description in revised manuscript.

Reviewer 3 Report

Dear Authors,

Thank You for submitting Your manuscript. Here are my comments:

  • I would suggest to include more details regarding primary treatment (transoral surgery vs compartmental surgery vs mono/bilateral neck dissection etc.) as well as regarding adjuvant treatment. It might be interesting to add a sub-analysis dividing patients according to the treatment received to describe the incidence of recurrences among each group;
  • The average follow up varied greatly among patients, as cases were recruited among a long period of time (2011-2020). I think this should at least be mentioned as a limitation of the study;
  • Please specify whether average or median values are used in Table 1 for continuous variables;
  • In general, I would suggest to include more details in the Results text (including %, HR, p, etc.) and to remove those from the Discussion paragraph;
  • No information on salvage treatments performed in relapsing patients is given;
  • I feel the discussion might be too focused on literature studies rather than Your results;
  • What were the reasons to recommend a yearly imaging during follow up for oral cancer? I think such a precise recommendation is not fully supported by Your results, but rather that late stage and older patients might benefit from a stricter radiological follow up.

Author Response

Reviewer 3

I would suggest to include more details regarding primary treatment (transoral surgery vs compartmental surgery vs mono/bilateral neck dissection etc.) as well as regarding adjuvant treatment. It might be interesting to add a sub-analysis dividing patients according to the treatment received to describe the incidence of recurrences among each group;

Reply: Thanks for your suggestion. We’ll add more description regarding primary treatment (including adjuvant treatment) in revised manuscript. We’ll add more subgroup analysis also.

The average follow-up varied greatly among patients, as cases were recruited among a long period of time (2011-2020). I think this should at least be mentioned as a limitation of the study;

Reply: Thanks for your reminding. We’ll add more description in the limitation section.

Please specify whether average or median values are used in Table 1 for continuous variables;

Reply: Sorry about that. Average values are used in Table 1 for continuous variables. We’ll add more description in revised manuscript.

In general, I would suggest to include more details in the Results text (including %, HR, p, etc.) and to remove those from the Discussion paragraph;

Reply: Thanks for your suggestion. As most of the detailed information already listed in the tables, we’ll remove all % and number from discussion section. In addition, we’ll move those that are not in the tables to the results section. (In the end of results section)

No information on salvage treatments performed in relapsing patients is given;

Reply: Thanks for your suggestion. We’ll add more description of salvage treatments in relapsing patients in revised manuscript. (In the end of results section)

I feel the discussion might be too focused on literature studies rather than Your results;

Reply: Thanks for your suggestion. We’ll add more description regarding our results in revised manuscript.

What were the reasons to recommend a yearly imaging during follow up for oral cancer? I think such a precise recommendation is not fully supported by Your results, but rather that late stage and older patients might benefit from a stricter radiological follow up.

Reply: Thanks for your suggestion. We’ll modify the sentence in the conclusion.

Round 2

Reviewer 1 Report

I have no further comments.